# Quercetin Alleviates Endoplasmic Reticulum Stress-Induced Apoptosis in Buffalo Ovarian Granulosa Cells

**DOI:** 10.3390/ani12060787

**Published:** 2022-03-20

**Authors:** Weihan Yang, Runfeng Liu, Qinqiang Sun, Xingchen Huang, Junjun Zhang, Liangfeng Huang, Pengfei Zhang, Ming Zhang, Qiang Fu

**Affiliations:** State Key Laboratory for Conservation and Utilization of Subtropical Agro-Bio Resources, College of Animal Science and Technology, Guangxi University, Nanning 530004, China; yangweihan2021@163.com (W.Y.); liurunfeng101@163.com (R.L.); gxsunqinqiang@163.com (Q.S.); hxcyyds6688@126.com (X.H.); zjjyyds6688@163.com (J.Z.); dwfzs101@163.com (L.H.); gxzhangpf@163.com (P.Z.)

**Keywords:** quercetin, endoplasmic reticulum stress, buffalo granulosa cells, apoptosis

## Abstract

**Simple Summary:**

Granulosa cells are critical components of the ovary that nurture germ cells and sustain oocyte maturation. Apoptosis of granulosa cells leads to follicular atresia, which in turn leads to female infertility. There are many reasons for the apoptosis of granulosa cells, one of which is apoptosis induced by endoplasmic reticulum stress. We found that quercetin could attenuate the effects of endoplasmic reticulum stress on granulosa cells by the PERK/CHOP signaling pathway. The results provide a novel strategy for inhibiting the apoptosis of granulosa cells.

**Abstract:**

Endoplasmic reticulum (ER) stress plays a crucial role in granulosa cell (GCs) apoptosis, which is the main cause of follicular atresia. Quercetin (QC), a plant-derived flavonoid, has antioxidant, anti-inflammatory, and other biological properties. However, whether QC can alleviate the effects of ER stress on buffalo GCs remains unknown. In this study, we constructed an ER stress model in buffalo GCs by using tunicamycin (TM) and pre-treated with QC to explore the effect of QC on cells under ER stress. Apoptosis was detected by Annexin fluorescein 5 isothiocyanate (V-FITC), and the expressions of mRNA and related proteins involved in ER stress and apoptosis were detected via real-time polymerase chain reaction and Western blot. The results revealed that ER stress can cause apoptosis in GCs, whereas QC pre-treatment can prevent apoptosis caused by ER stress. After pre-treatment with QC, the expression levels of ER stress-related genes and proteins significantly decreased, pro-apoptotic genes were significantly down-regulated, and anti-apoptotic genes were significantly up-regulated. Furthermore, the results of Chop gene overexpression suggested that QC alleviated ER stress via the PERK/CHOP signaling pathway. In this study, we preliminarily elucidated that QC alleviates ER stress-induced apoptosis in buffalo GCs, and the results suggest a novel strategy for delaying follicular atresia by inhibiting GCs apoptosis.

## 1. Introduction

The endoplasmic reticulum (ER) is an important organelle with a complex tubular membrane exclusively present in the cytoplasm of the eukaryotic cells. It serves as an intracellular calcium reservoir that regulates and maintains intracellular calcium balance. It is also involved in protein synthesis, folding, transport, and modification [1,2,3,4]. Therefore, ER homeostasis is critical for maintaining normal physiological activities of the eukaryotic cell. ER dysfunction disrupts protein homeostasis in cells leading to the activation of unfolded protein response (UPR), which is crucial for cell survival [5,6,7]. UPR promotes ER degradation of functionally defective or unfolded proteins while maintaining ER homeostasis. Early UPR is the organic self-compensation process that protects cells. If this imbalance exceeds the body’s self-regulating ability, cell apoptosis will eventually occur [8,9].

Recent studies have indicated that the ER stress plays a crucial role in the regulation of apoptosis. In contrast to the death receptor and the mitochondrial pathway [10], the mechanism of ER stress-induced apoptosis is mainly mediated by the protein kinase RNA-activated-like ER kinase (PERK)/C/EBP homologous protein (CHOP) and inositol-requiring enzyme 1a (IRE1)/c-Jun N-terminal kinase (JUK) pathways [11]. PERK enhances the expression of activating transcription factor 4 (ATF4). Upregulated ATF4 protein activates the ability of ER to process unfolded proteins and release ER stress. ATF4 expression promotes the expression of CHOP, causing downstream proteins such as B cell lymphoma/leukaemia-2 (BCL-2) and BCL-2 associated protein X (BAX), which induces apoptosis. IRE1 activates the JNK signaling pathway, which prompts CHOP to activate downstream apoptosis-related genes. Previous studies have confirmed that ER stress induces apoptosis of ovarian granulosa cells (GCs) in mice [12]. Apoptosis of GCs is the primary cause of follicular atresia, which causes in female infertility [13,14]. In domestic animal breeding, ER stress greatly restricts the utilization of superior breeds in production, causing the waste of superior female resources [15]. Therefore, elucidating the mechanism of ER stress and inhibiting apoptosis in GCs is critical for animal reproduction.

Quercetin (QC), also known as QC flavonoid, is abundantly found in fruits and vegetables. QC exhibits antioxidant, anti-inflammatory, antiviral, and other biological activities in humans, mice, and other model animals [16]. In rats, QC protected against cisplatin-induced ovarian toxicity, improved ovarian tissue lesions, and increased the percentage of healthy follicles. QC can also act as a metal chelator and ROS scavenger to prevent changes in bovine semen quality and maintain normal reproductive function of semen [17,18]. Both in vivo and in vitro studies have shown that QC can potentially regulate numerous molecules and protect cells from harm through various mechanisms. However, it is unclear whether QC can alleviate the effect of ER stress on buffalo GCs. Hence, in view of the significance of ER stress in GCs dysfunction, we aimed to ascertain the protective nature of QC against ER stress. For this purpose, we constructed an in vitro ER stress model to explore whether QC can alleviate the effect of ER stress on cells impact.

Buffalo, as an important livestock, is adapted to the humid tropical climate of the south. Guangxi is a large area for buffalo breeding. As a new economic growth point in southern China, buffalo breeding is the backbone of rural economic development [19]. However, the reproductive efficiency of buffalo is lower than that of other animals, which has a great relationship with the oocytes and GCs in the ovary. Here, we hypothesize that ER stress-induced apoptosis in GCs and QC supplementation have a positive effect on buffalo GCs by the PERK/CHOP signaling pathway. To validated this hypothesis, we explored the effect of ER stress on buffalo GCs and the answer to whether QC could alleviate the effect of ER stress on buffalo GCs. It provides some new ideas for relieving follicular atresia and improving the reproductive capacity of female animals.

## 2. Materials and Methods

### 2.1. Cell Culture

Buffalo (10 individuals, age between 24 and 36 months at post puberty) ovaries were collected from a local abattoir (Nanning, China) in November to January. The ovaries were shipped immediately to the laboratory within 2 h in a thermos (36–37 °C), following the Wang et al. method with appropriate adjustments [20]. Twenty buffalo ovaries were taken and washed using preheated isotonic sterile saline solution. The follicle fluid was collected using a disposable syringe and centrifuged at 1500 rpm for 5 min, after which the precipitate was collected and washed thrice with Dulbecco’s phosphate-buffered saline (D-PBS) (Thermo Fisher Scientific, Rockville, MD, USA). Subsequently, the precipitate was filtered twice using a 40 μm cell strainer to remove the cell debris. The purified GCs were resuspended in Dulbecco’s modified Eagle’s (DMEM) medium (Gibco, Gaithersburg, MD, USA) supplemented with 10% fetal bovine serum (Gibco, Waltham, MA, USA), 100 U/mL penicillin and 100 μg/mL streptomycin (Hyclone, Logan, UT, USA). The cells were then cultured at 37 °C in a 5% CO_2_ incubator with maximum humidity. After cell differentiation, the model of cell apoptosis induced by ER stress was established as described by Suganya et al. [21]. Briefly, the culture medium was removed and replaced with the culture medium consisting of diluted tunicamycin (TM) (Solarbio, Beijing, China). GCs were then subjected to increasing doses of TM at concentrations of 1, 1.5, 2, 2.5, and 5 μg/mL and cultured at 37 °C for approximately 24–48 h in a 5% CO_2_ incubator.

### 2.2. Quercetin Treatment of GCs

Approximately 2000 GCs in 100 μL solution were added to 96-well plates. After the cells grew to adhere to the wall completely, QC (Yuanye, Shanghai, China) was added to the culture medium to maintain the final concentrations of 1, 1.5, 2, 2.5, 5, 10, and 20 μg/mL. The cells were cultured at 37 °C for 24 h, 48 h, and 72 h. Cell viability was detected using the cell counting kit-8 (CCK-8) (Beyotime, Shanghai, China), according to the instructions of the manufacturer. Ten microliters of CCK-8 reagent was added to the cells, and the cells were incubated at 37 °C for 1 h. Cell viability was measured using the absorbance wavelength of 450 nm. The buffalo GCs were pre-treated in four different groups, which were the control, TM (2.5 μg/mL), QC (2 μg/mL), and TM (2.5 μg/mL) + QC (2 μg/mL) groups. The cells in each group were cultured at 37 °C with 5% CO_2_.

### 2.3. Cell Immunofluorescence Analysis

The adherent cells were rinsed with D-PBS before fixing them with 4% paraformaldehyde (Solarbio, Beijing, China) for 15 min at RT. The cells were permeabilized in 0.5% Triton X-100 (Solarbio, Beijing, China) for 20 min at RT. The cells were incubated with the follicle-stimulating hormone receptors (FSHR) primary antibody (Boster, Beijing, China) overnight at 4 °C. After the cells were washed twice using D-PBS, a fluorescent secondary antibody (ZSGB-Bio, Beijing, China) was added and incubated at RT for 1 h. Simultaneously, 4’, 6-diamidino-2-phenylindole (DAPI) (Sigma-Aldrich, St. Louis, MO, USA) was added and incubated for 5 min. The cells were observed and visualized using a fluorescence microscope (Nikon Eclipse C1, Tokyo, Japan).

### 2.4. Flow Cytometric Analysis

The Annexin-FITC/PI apoptosis kit (Beyotime, Shanghai, China) was used to evaluate the apoptosis of GCs according to the manufacturer’s instructions to detect cell apoptosis. The adherent cells were digested by adding trypsin. The lysed GCs were centrifuged at 1000 rpm for 5 min and were adjusted to a concentration of 1 × 10^6^ cells/mL. In the GCs, 5 μL Annexin V-FITC reagent and 10 μL propidium iodide (PI) were added. Early and later apoptotic cells were processed as a negative and positive control group, respectively. Flow cytometric analysis was performed using a flow cytometer (C6, BD, Franklin, TN, USA) and data were analyzed using the FlowJo software.

### 2.5. ROS Activity Detection

According to the instructions of the manufacturer. The supernatant of the treated cells was removed by aspiration and washed with D-PBS. A serum-free culture medium was used to dilute 2’,7’dichlorofluorescin diacetate (DCFH-DA) at a ratio of 1:1000 to make the final concentration 10 μmol/L, the diluted DCFH-DA was added, and then incubated in a 37 °C cell incubator for 20 min. The cells were washed with D-PBS to remove DCFH-DA that did not penetrate the cells, and the fluorescence intensity was observed under a fluorescence microscope.

### 2.6. Quantitative Real-Time PCR

Total RNA of the GCs was extracted by using TRIzol reagent (Thermo Fisher Scientific, Rockville, MD, USA) according to the instructions of the manufacturer. First-strand cDNA was synthesized using a SuperScript III reverse transcriptase master kit (Takara, Beijing, China) in a total volume of 20 μL. A quantitative real-time polymerase chain reaction (q-PCR) was performed to amplify all targeted genes on a LightCycler 480 real-time PCR instrument (Roche, Basel, Switzerland) using TB Green^®^ Premix Ex Taq^TM^ II Kit (Takara, Beijing, China) according to the manufacturer’s instructions. The q-PCR procedure was as follows: 95 °C for 1 min, then 40 cycles of 95 °C for 30 s, 95 °C for 5 s, and 60 °C for 30 s. GAPDH was used as an internal control. The relative expression levels from three experiments were analyzed using the 2^−ΔΔCT^ value. The primers sequences for q-PCR are listed in Appendix A.

### 2.7. Western Blot Analysis

The protein expression levels were validated via Western blot analysis. Proteins from GCs were extracted using lysis buffer 0.1% sodium dodecyl sulphate (SDS) and 1 mM phenylmethylsulphonyl fluoride (PMSF). Protein concentrations were measured using a bicinchoninic acid (BCA) protein assay kit (Beyotime, Shanghai, China) as per the manufacturer’s instructions. Aliquots of 30 μg of total protein were separated on 12% SDS-polyacrylamide gels. Furthermore, the protein was transferred onto polyvinylidene fluoride (PVDF) membranes using a Bio-Rad protein fast transfer membrane instrument (Hercules, CA, USA) at 0.2 A for 30 min. The membranes were incubated overnight at 4 °C with primary antibodies (Abcam, Cambridge, UK). The membranes were further incubated for 1 h at RT with a secondary antibody conjugated with horseradish peroxidase The chemiluminescent signal was detected using the (5-bromo-4-chloro-3-indolyl-phosphate/nitro blue tetrazolium) BCIP/NBT detection reagents (Beyotime, Shanghai, China).

### 2.8. Construction of a Plasmid

Primers were designed according to the *Chop* gene sequence provided by National Center for Biotechnology Information. *EcoRI/XhoI* restriction enzymatic digestion sites were added at both ends of the primers to amplify the cDNA sequence containing the *Chop* open reading frame. The amplification product was inserted into the plasmid for generating pcDNA3.1-CHOP. The sequence was confirmed by DNA sequencing.

### 2.9. Cell Transfection

Buffalo GCs were cultured in a DMEM medium containing 10% FBS and 1% Penicillin–streptomycin double-antibody in a 5% CO_2_ incubator. The cells were cultured in 6-well plates. When the cell density reached approximately 70–90%, the recombined plasmid was cotransfected into GCs using Lipofectamine R3000 reagent (Thermo Fisher scientific, Rockville, MD, USA). According to the instructions of the manufacturer. Dilute Lipofectamine R3000 and P3000 reagents using Opti-MEM medium. After dissolving 4 μg of plasmid in the P3000 reagent, the Lipofectamine R3000 reagent was added to the cells and incubated for 15 min at RT. Liposome complexes were transfected into GCs. Simultaneously, the negative control was transfected with an empty plasmid. After a 12 h incubation period, 2 μg/mL puromycin was added for GCs selection.

### 2.10. Statistical Analysis

All data were analyzed using SPSS 13.0 software, all treatments were performed in three independent biological replicates and three technical replicates, data were expressed as means ± standard deviation, and statistical analysis was performed using one-way ANOVA. Bonferroni multiple comparison tests were performed. A statistically significant difference was defined as *p* < 0.05.

## 3. Results

### 3.1. GCs Culture

The buffalo GCs were observed under a stereomicroscope and revealed a single-layer adhering growth. After 48 h of culture, the cells exhibited complete adherent growth with strong growth ability, rapid proliferation, large cell volume, clear outline, and long spindle shape (Figure 1A). After 4 days of culture, the cells aggregated into a single cell layer and covered the bottom of the culture dish. Immunofluorescence analysis revealed that the specific antibody of FSHR receptors was detected in GCs. Subcellular location analysis revealed that the FSHR protein was primarily expressed in the cell nucleus and membrane (Figure 1B).

### 3.2. Construction of an ER Stress Model

The TM-induced strategy was used to construct an ER stress model of buffalo GCs. The cell apoptosis was evaluated via flow cytometry after staining the cells with Annexin V-FITC and PI. The results indicated that TM increased the apoptosis of buffalo GCs (*p* < 0.001) in a time- and concentration-dependent manner (Figure 2A). TM treatment at a concentration of 2.5 μg/mL for 48 h increased cell apoptosis by up to 34% in buffalo GCs, and hence can be considered an appropriate strategy to construct an ER stress model. The mRNA expression levels of ER stress marker genes, such as *Chop*, *Perk*, *Atf4*, *Atf6*, and *Ire1*, were detected via q-PCR. Figure 2B illustrated that the marker genes were highly expressed in TM-induced GCs. Moreover, according to the Western blot densitometric readings (Appendix A), analysis revealed that the expression levels of ATF4 and CHOP protein were consistent with mRNA expression levels, and that BAX and BCL-2 expression levels were consistent with apoptosis data (Figure 2C). These results demonstrated that TM promotes apoptosis through ER stress, and a TM-induced ER stress model of buffalo GCs was successfully constructed.

### 3.3. Effect of QC on TM-Induced Apoptosis in Buffalo GCs

To evaluate the protective effects of QC on TM-induced apoptosis, QC at different concentrations (1, 1.5, 2, 2.5, 5, 10, and 20 μg/mL) was incubated with the cells for 24, 48, and 72 h. We observed that QC promoted cell vitality and cell proliferation in a concentration-dependent manner in buffalo GCs. As presented in Figure 3A, cell viability and proliferation after QC pre-treatment for 48 h were higher than those after 24 h and 72 h, The QC group was compared with the control group. Further comparison of QC concentration revealed that QC concentration-dependent increase in cell vitality varied from 130% to 150%, with QC pre-treatment at a concentration of 2 μg/mL having the strongest effect on promoting cell proliferation (*p* < 0.01). However, a high concentration of QC (approximately 10–20 μg/mL) suppressed the growth of GCs and decreased cell vitality.

Furthermore, to validate that QC exerted protective functions by suppressing TM-induced apoptosis in buffalo GCs, the cells were pre-treated with 2 μg/mL of QC for 8 h and subjected to TM stimulation. As presented in Figure 3B,C, the apoptotic ratio of GCs reached 30% after TM stimulation. Remarkably, the apoptotic ratio decreased with QC pre-treatment at a concentration of 2 μg/mL (*p* < 0.01). The intensity of intracellular reactive oxygen species (ROS) after QC pre-treatment was lower than that in the TM treatment group (*p* < 0.01) (Figure 3D,E). These results demonstrated that the protective effect of QC on TM-induced apoptosis alleviated apoptosis in buffalo GCs.

### 3.4. Effect of QC on Gene Expression in GCs

To confirm the effects of QC on TM-induced ER stress in buffalo GCs, the expression levels of *Bax*, *Caspase3*, *Caspase9*, and *Bcl**-2* genes were evaluated via q-PCR. TM can promote apoptosis of buffalo GCs by upregulating the expression of *Bax*, *caspase3*, and *caspase9* (Figure 4A), confirming that a higher apoptotic percentage is associated with gene regulation of ER stress. However, QC pre-treatment reduced the expression levels of these genes to normal levels of the control group. Moreover, the anti-apoptotic *Bcl**-2* gene was expressed at a lower level in TM-induced GCs, which increased following QC pre-treatment *(**p* < 0.05) (Figure 4A). According to the Western blot densitometric readings (Appendix A), analysis revealed that BAX, Caspase9, and BCL-2 proteins were detected in GCs (Figure 4B). The protein expression profiles were consistent with mRNA expression levels.

ER stress-related genes, *Atf4*, *perk*, *Chop*, and *Ei**f2a*, were evaluated to validate the effect of ER stress during QC pre-treatment in buffalo GCs. As presented in Figure 4B, four ER stress-related genes were upregulated after TM stimulation and were positively correlated with apoptosis. The expression level of these genes was decreased after QC pre-treatment in buffalo GCs (*p* < 0.01). These results illustrated that QC pre-treatment suppressed apoptosis induced by ER stress. Moreover, the mRNA and protein expression of *Atf4* and *Chop* genes in the QC pre-treatment group were lower than those in the TM-inducing group in buffalo GCs (*p* < 0.01) (Figure 4B,C), suggesting that QC suppresses ER stress by activating the PERK/CHOP signaling pathways.

### 3.5. Activation of the PERK/CHOP Pathway

The PERK/CHOP signaling pathway is one of the pathways through which ER stress promotes apoptosis. Further analysis was performed to evaluate the relationship between the PERK/CHOP signaling pathway and QC pre-treatment in buffalo GCs. As presented in Figure 5, the expression level of the *Chop* gene increased in buffalo GCs after plasmid transfection, indicating that the *pcDNA3.1-**CHOP* recombination plasmid was successfully constructed for activation of the Chop gene (Figure 5A). Flow cytometric analysis revealed that activation of the *Chop* gene stimulated buffalo GCs to increase apoptosis (Figure 5B,C). Simultaneously, intracellular ROS activity was also enhanced (Figure 5D,E). However, the transfected buffalo GCs pre-treated with QC retained the normal expression level of the *Chop* gene. Cell apoptosis decreased in buffalo GCs with QC pre-treatment (*p <* 0.01); although, *Chop* gene expression was activated by plasmid transfection, QC pre-treatment reduced ROS activity (*p* < 0.01) (Figure 5B,C). These findings indicated that QC pre-treatment could inhibit the PERK/CHOP signaling pathway, alleviating ER stress.

## 4. Discussion

Despite the rising prevalence of cell apoptosis, little study has directly addressed the relationship between ER stress and GCs apoptosis. It is unclear whether QC can alleviate the effect of ER stress on buffalo GCs. In this study, we hypothesized that ER stress probably induces apoptosis in buffalo GCs, and QC has the protective nature against ER stress. For this objective, we constructed an in vitro ER stress model and further investigated the functions of QC in buffalo GCs, along with the molecular mechanisms of action.

The results demonstrated that TM-induced ER stress induces apoptosis in buffalo GCs. TM, a nucleotide sugar analog, is a naturally occurring antibiotic mainly produced by the Streptomyces species. TM inhibits protein glycosylation, causing protein misfolding and UPR activation, leading to apoptosis [21,22]. A previous study found that TM could induce *Chop* gene expression in endothelial cells, the expression of *Grp78* gene was significantly elevated, and the expression of downstream *Bax* apoptotic genes was activated by the *Chop* gene [21]. Therefore, we hypothesized that TM induced apoptosis in buffalo GCs in a similar pattern. In our study, flow cytometry confirmed that TM promoted apoptosis of buffalo GCs, and the apoptosis-related genes *Bax*, *Caspase3*, and *Caspase9* were expressed at higher levels after TM treatment. However, the expression level of the anti-apoptotic gene *Bcl-2* was down-regulated. Furthermore, the expression of ER stress-related genes such as *Perk*, *Chop*, *Atf4*, *Atf6*, and *Ire1* was significantly increased after TM treatment. The PERK signaling pathway activates the expression of the downstream CHOP protein and promotes apoptosis. These findings confirm that TM induces apoptosis in buffalo GCs via ER stress.

QC is the most ubiquitous of dietary flavonoids and has been reported to be a potent antioxidant that inhibits apoptosis through anti-inflammatory, antioxidant, and other pathways. Numerous beneficial effects, in vivo and in vitro, of QC have been reviewed [23]. In clinical settings, QC is used to treat a variety of diseases such as bacterial infections, hypertension, and neurodegenerative diseases. In animal experiments, dietary supplementation of QC can promote follicular growth and increase GCs viability in heat-stressed rabbits [24]. QC has estrogen-like activities; thus, it increased the follicle population in the ovaries of mice [25]. In this study, QC treatment at various concentrations was examined according to previous description [26]. Similar anti-apoptotic effects were observed, and low concentrations of QC (1–2 μg/mL) can promote cell proliferation and improve cell viability, which is consistent with the results of previous studies [27]. Moreover, the effect of QC on TM-induced apoptosis in GCs was investigated, and the findings demonstrated that QC considerably decreased apoptosis [28,29]. Suganya et al. [21] demonstrated that QC pre-treatment of endothelial cells with TM-induced apoptosis reduced *Bax* gene expression while increasing anti-apoptotic *Bcl-2* gene expression, suggesting the potential of QC to prevent apoptosis. In this study, we observed the concentration-dependent cytoprotective effects of QC on TM-induced toxicity in buffalo GCs. The mRNA and protein expression levels of the *Bax* gene were decreased and the anti-apoptotic *Bcl-2* gene was increased. Excessive reactive oxygen species (ROS) production leads to increased mitochondrial permeability, which releases cytochrome c from the inner membrane space into the cytoplasm and activates a cascade of caspases, a family of caspases that are responsible for apoptosis. As a key factor in the execution of apoptosis, it initiates downstream apoptosis; ultimately, it leads to typical biochemical and morphological changes. In this study, we observed that the gene expression levels of caspase3 and caspase9 were decreased and ROS activity was also suppressed after QC pre-treatment. The results of gene expression confirmed the hypothesis that QC alleviated TM-induced apoptosis in GCs.

However, the contribution of QC to ER stress remains unclear. QC induced ER stress in cancer cells but inhibited ER stress when normal cells encountered ischemia/reperfusion injury [30]. A detailed study on the mechanisms of QC will help in better understanding the functions. Based on the previous observations of QC inhibiting glucosamine-induced apoptosis at high concentrations [26], we speculated that QC may alleviate apoptosis by inhibiting ER stress in GCs. The expression profiles of ER stress-related genes were further assessed to give a better understanding of the effect of ER stress. We observed that the expression levels of ER stress markers such as *Atf4*, *Perk*, *EIF2a*, and *Chop* genes were decreased, confirming that QC treatment suppressed ER stress in GCs. These findings were consistent with those of previous studies in endothelial cells [26].

Apoptosis is induced by three typical signaling pathways in response to ER stress, which are as follows: PERK–EIF2α–ATF4–CHOP, IRE1α–XBP1, and ATF6 [4]. It is well-known that the phosphorylation of PERK serves as an initiator for its activation status. EIF2α serves as an effective downstream factor for PERK stimulation, and the expression of ATF4 is enhanced during this process. ATF4 drives the activation of the proapoptotic transcription factor CHOP [31]. The mRNA and protein expression levels of the PERK/CHOP signaling pathway were significantly decreased in this study, suggesting that inhibiting the PERK/CHOP signaling pathway alleviates cell apoptosis. Several studies have demonstrated that activating *Chop* can intensify ER stress and induce apoptosis. Therefore, a pcDNA-3.1-CHOP overexpression plasmid was constructed in this study. The *Chop* gene activation caused in a significant increase in apoptosis. However, the apoptotic ratio was decreased after QC pretreatment. This study provides concrete evidence that the PERK/CHOP axis of the ER stress response contributes to TM-induced apoptosis in GCs, and QC may inhibit ER stress and alleviate apoptosis through suppressing the PERK/CHOP signaling pathway.

This study constructed an ER stress model in GCs using TM, demonstrating that ER stress induces apoptosis in buffalo GCs. In vitro experiments in ER stress GCs model confirmed our hypothesis that QC attenuated apoptosis of GCs. By constructing the pcDNA3.1-CHOP overexpression plasmid, the mechanism of QC in alleviating ER stress has been elucidated, which is associated with the inhibition of ER stress and the suppression of the PERK/CHOP signaling pathway. Our results suggest that QC is a promising approach for reducing apoptosis in buffalo GCs. However, the specific mechanism of action of QC to relieve ER stress and in vivo experiments need to be further studied.

## 5. Conclusions

In conclusion, QC can attenuate apoptosis in buffalo GCs by inhibition of ER stress. Moreover, the PERK/CHOP signaling pathway may be a key pathway of QC in alleviating ER stress and reducing apoptosis. QC treatment is a novel strategy for inhibiting apoptosis in buffalo GCs.

## Figures and Tables

**Figure 1 animals-12-00787-f001:**
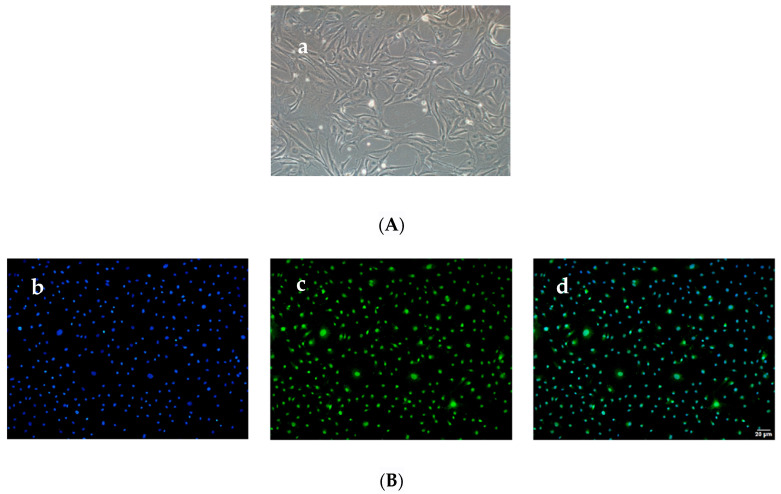
Buffalo GCs culture and identification: (**A**) (**a**): The morphology of buffalo GCs was observed under the microscope. (**B**) Identification of buffalo GCs by cellular immunofluorescence. (**b**): Cell nucleus were stained by DAPI (blue), (**c**): Cell nucleus and cell membranes were stained by FSHR (green), (**d**): Merge of (**b**,**c**).

**Figure 2 animals-12-00787-f002:**
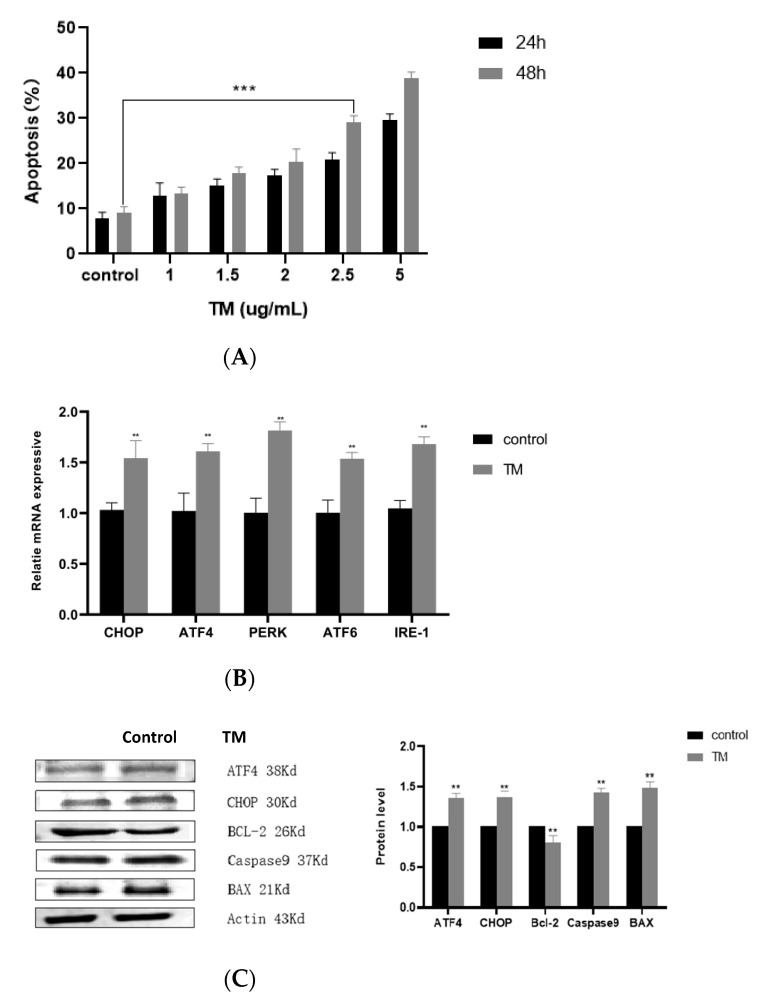
Construction of an ER stress model: (**A**) Cells were treated with TM at the indicated dose of 0–5 µg/mL for 24 h and 48 h. Ten thousand cells were treated with Annexin V-FITC and PI reagent and analyzed on a fluorescence-activated cell sorting (FACS) instrument. Percentages of Annexin V-positive/PI-negative and Annexin V-positive/PI-positive cells are shown. (**B**) q-PCR analysis of *Perk*, *Ire1*, *Atf6*, *Chop*, and *Atf4* mRNA expression in GCs after TM treatment. (**C**) Western blotting was performed to detect the protein level with the indicated antibodies against ATF4, CHOP, BCL-2, Caspase9, and BAX. The asterisks refer to the level of significance (** *p* < 0.01, **** p* < 0.001). Values are presented as means ± SD for 3 biological replicates.

**Figure 3 animals-12-00787-f003:**
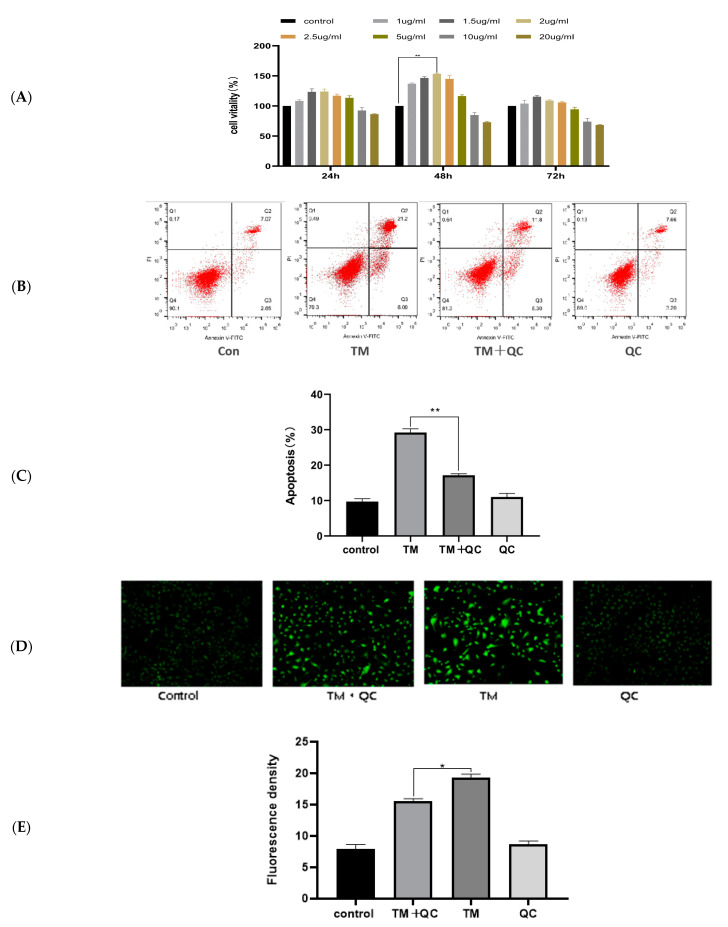
The effect of QC on TM-induced apoptosis in GCs. control: control group; TM + QC: pre-treated with QC, and then treated with TM; TM: TM alone treatment group; QC: QC alone treatment group: (**A**) The effects of different concentrations of QC treatment on cell viability for 24, 48, and 72 h. (**B**) The GCs were pretreated with QC for 8 h, and then treated with 2.5 µg/mL TM for 40 h, and the apoptosis rate was detected by flow cytometry. (**C**) Statistics on apoptosis. (**D**) The intensity of intracellular ROS was examined using fluorescence microscopy. The higher the fluorescence intensity, the stronger the intracellular ROS activity. (**E**) Fluorescence Intensity Statistics. The asterisks refer to the level of significance (* *p <* 0.05, ** *p*  *<*  0.01).Values are presented as means ± SD for 3 biological replicates.

**Figure 4 animals-12-00787-f004:**
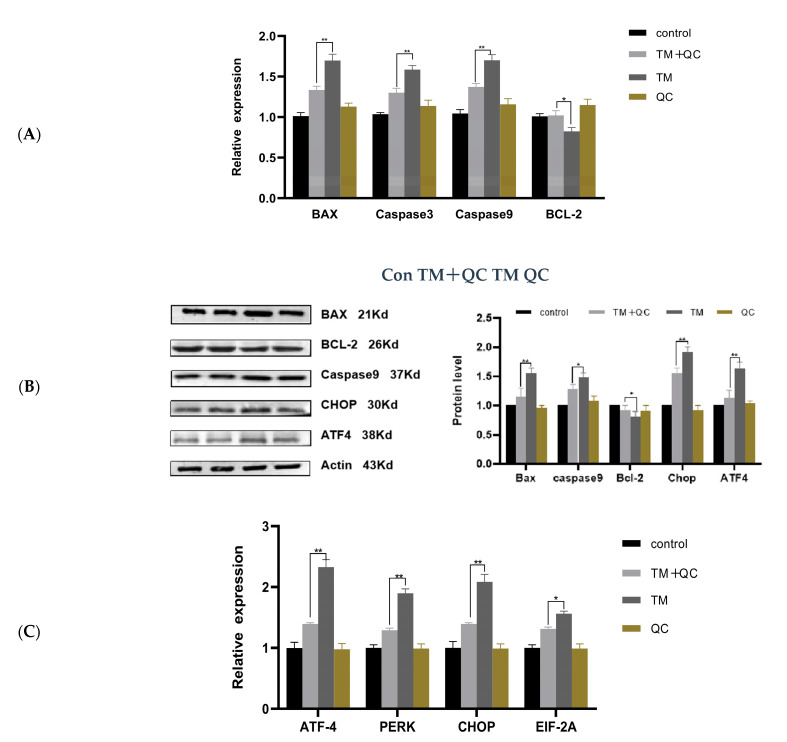
Gene expression changes in GCs after QC treatment: (**A**) The mRNA expression level of *Bax*, *Bcl2*, *Caspase3*, and *Caspase9* in different treatment groups was examined by q-PCR. (**B**) The expression levels of proteins in different treatment groups were detected by Western blotting. (**C**) The mRNA expression of *Atf4*, *Chop*, *Perk*, and *Eif2a* in different treatment groups was examined by q-PCR. The asterisks refer to the level of significance (* *p* < 0.05, ** *p* < 0.01). Values are presented as means ± SD for 3 biological replicates.

**Figure 5 animals-12-00787-f005:**
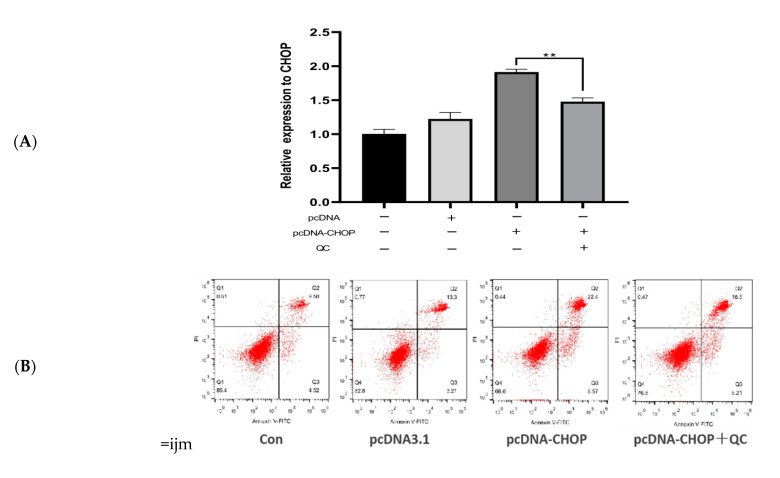
Effects of activation of Chop gene. control: blank control group; pcDNA3.1: transfected with pcDNA3.1 plasmid group; pcDNA3.1-CHOP: transfected with Chop overexpression plasmid group; pcDNA3.1-CHOP + QC: pretreated with QC, then transfected with Chop overexpression plasmid group: (**A**) The expression level of the Chop gene after transfection of pcDNA3.1-CHOP plasmid. (**B**) After transfection of pcDNA-CHOP plasmid, cell apoptosis was evaluated by flow cytometry. (**C**) Statistics of apoptotic cells. (**D**) After transfection of pcDNA-CHOP plasmid, the intensity of ROS in cells was detected by fluorescence microscope. The higher the fluorescence intensity, the higher the intracellular ROS content. (**E**) Fluorescence density statistics. The asterisks refer to the level of significance (** *p <* 0.01). Values are presented as means ± SD for 3 biological replicates.

## Data Availability

The data presented in this study are available on request from the corresponding author. The data are not publicly available due to privacy restrictions.

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
