# Peer review of "Quercetin Alleviates Endoplasmic Reticulum Stress-Induced Apoptosis in Buffalo Ovarian Granulosa Cells"

_animals, 2022, doi:10.3390/ani12060787_

Round 1

Reviewer 1 Report

In general, the article is correctly prepared with very clear results obtained and well-summarized conclusions.

But I have observed some errors that should be corrected, mainly in the material and methods and results.

Introduction:

Lines 44 to 46: After Kim et al., 2008 there is a full stop that shouldn't be there for the sentence to make sense. Please correct it.

There is no clearly defined goal. In lines 77 to 79 what is written seems more like a conclusion than an objective marked in this work. Please rewrite or modify it.

Material and methods:

In general, it is necessary to specify much more all the reagents used since the information given is, most of the time, very diffuse. This makes it very difficult if another researcher wants to repeat your experiments due to the lack of information provided. For example, in line 93 the culture medium used, DMEM, from Gibco but it should have at least the catalog number. The same occurs for most references. It is necessary that everything is well described so that it can be reproduced by any other researcher.

In section 2.10, on line 166 it says “…1% double antibiotics in a 5-5 CO2…” but, which two antibiotics are they? You should specify them.

Results:

3.1 GCs culture:

The image in Figure 1A is clear but there is no way of knowing which cells belong to the granulosa cells. I think it would be interesting to carry out a specific staining in bright field to confirm that they are that specific cell type (especially to the reader of the article) using, for example, ER beta antibody.

In line 188, when it says “FSHR protein was primarily expressed in the cell nucleus and membrane (Figure 1B)”, observing the image it is not possible to see the expression in the membrane. It may be because the image magnification is not enough. If this is the case, please change the image to a more representative one or else what is being said would not be correct.

The caption of figure 1 gives little descriptive information about what it represents. Also, on line 195 it says DPAI when it should say DAPI. Please correct it.

3.2 Construction of an ER stress model.

In line 201 where it says "... at a concentration of 2.5 μg/mL for 48 h reduced cell apoptosis by up to 34% in buffalo...", shouldn't it be "increased" as shown in Figure 2A?

Figure 2C shows several western blots of various proteins, but the differences between Control and TM are not well observed, so I think they should be improved. Furthermore, the "Con" and "TM" elements are not centred with the bands. Also, the molecular weights are indicated as Ku and not as Kd. I am sure it is because of a computer bugs but please correct them.

3.3 Effect of QC on tm-induced…

In this section a Figure 3 is shown but the results obtained are not detailed in the section with the images, tables or diagrams that are in Figure 3. In addition, the caption of Figure 3 provides very little information and little description of what is observed making it of little use. Please fix these bugs.

3.4 Effect of QC on gene expression…

In this section, I would change the order in figure 4, putting figure 4C as 4B and vice versa since it is presented in the text following that order. Thus, the order of appearance of the different figures would be A, B and C and not as it is now which is A, C and B. This is a suggestion.

In general, in the figures you should check that there are no errors in the units of measurement and that the labels are centred with the corresponding image.

In addition, a revision of the text would be advisable since some typographical errors such as periods, commas (line 336) or errors in words (line 350) are observed that must be corrected.

Author Response

请参阅附件

Reviewer 2 Report

ANIMALS - MDPI; Referee’s Evaluation Report

Quercetin alleviates endoplasmic reticulum stress-induced apoptosis in

buffalo ovarian granulosa cells (ORIGINAL ARTICLE)

Comments to Authors/Editor:

The paper of Yang & colleagues aimed to understand the protective mechanism of quercetin, -a plant-derived flavonoid with anti-oxidant and anti-inflammatory activities-, as a reliever of endoplasmic-reticulum stress, an important cause of granulosa cell apoptosis and, in turn, follicular atresia. This manuscript falls within the scope of ANIMALS. The manuscript is sufficiently informative for the replication of the study.  In general, the organization of the experiment seems to be well designed, yet, the English quality, grammar, and sentence structure must be greatly improved; L45, non-sense sentence; correct accordingly. Besides, the authors are not following the requested format by the MDPI journals; please cite with numbers, not with last names. No Simple Summary was included. L47; define PERK/CHOP & IRE1/JUK pathways. Every sentence must be supported by the proper citation; correct along with the entire manuscript. L54; “and a unique mechanism????, correct accordingly. L44 to L77; please rewrite these paragraphs; their actual format is extremely confusing.  The authors must define any used abbreviation the first time appearing in the manuscript.  L73 to 79; not a single citation was included. The authors must include information regarding buffaloes; the world inventory, the Asian inventory, and China´s buffalo inventory along with productive, economic, and social importance in China. The Introduction section is extremely long; it must be shortened.  While the objectives of the study were stated, no working hypothesis of the study was proposed; this is a must. Please include the contribution of buffaloes to the national livestock sector; why do the authors use female buffaloes instead of cows, sheep, or goats??? Are buffaloes important from a productive, economic or social standpoint???  The Introduction section is quite long; are the authors comfortable with 79 lines as an Introduction?? Some information must be moved to the Discussion section. In the Material & Methods section, I strongly recommend including a figure with the actual experimental protocol across time (i.e. a timeline of actions); this is a must. Besides, the authors must indicate how the experimental samples were managed during the experimental period. Both, in the Abstract and M&M sections, the authors must include where the experiment was carried out (NL, WL), including the months when the samples were collected, and define if buffaloes depict a seasonal or continuous reproduction. Please state, if is the case, whether the experimental samples were collected at the beginning, the middle, or the last stage of the breeding season.  In the Introduction section, every statement must include at least one reference; it is real nonsense not to include the required bibliographical sources in any scientific report; correct accordingly along with the manuscript. In the M&M section, the authors must define the number of replicates used in this study; were the samples collected from homogeneous experimental units??? That is, the age, body weight, body condition score, lactation number; please carefully explain the physiological nature of the collected samples. L87 to 98; again, not a single citation was included in order to support the methods used; this is a must.  The same is true regarding L99 to 109, L110 to 119, L120 to 128, L129 to 135, L136 to L145, L146 to L163; are the authors comfortable with this approach?? In general, the materials, standards, and methods used are relevant and in accordance with the objectives of the study. Also, all the sampling techniques, the reported methodology, and consider techniques used in the experiment are detailed and accurate, while in accordance with the objectives of the study, yet, without any single citation. The English quality must be improved also in this section. The authors must clearly define L174 to 178; statistical analysis or statistical analyses???; the authors must include the statistical model used along with the analyses. Since the authors used increased tunicamycin concentrations, the authors must explain why they used a one-way analysis of variance to detect any treatment effect.  In this section, the authors must be very careful to clearly define the number of replicates per TM & QC concentration as well as the sample size in the control group. Certainly, based on the previous comment, the authors must explain why they are reporting means instead of least-square means? The experimental design, and statistical models, were not described well enough for the reader to understand how the experiment was carried out. Besides, the authors must describe if the response variables depicted a normal distribution or if they required an adjustment or transformation in order to perform the analyses through ANOVA. In the Results section; L199, significantly???, at P<0.01 or P<0.05????; do not use the word “significantly”, use the probability level itself. Regarding the Results section, the novelty value of the results is reasonable, yet, the authors must include some kind of quantitative information aside from the P-values of the observed results regarding the response variables.  Also, if no differences occurred among experimental groups for a defined variable, the authors must include the average value for such response variables observed in the study. About the Discussion section, at the beginning of the Discussion, I do strongly suggest initiating this section including the working hypothesis of the study. Authors must define if, with the obtained results, such a hypothesis is rejected or non-rejected. For this reason, the authors must include the working hypothesis prior to the objectives in the Introduction section. In addition, the authors must follow the same order in this section according to that proposed in the Results section; the authors must homogenate the presentation of the Results and Discussion.  The authors must link, in a logical fashion, their main findings along with the discussion section, to compare & discuss and, thereafter, be able to propose some physiologic explanations for such specific outcomes, considering previous similar studies from the scientific literature. In general, the authors made an accurate interpretation of the main findings.  The Discussion section is quite extended; the authors must focus their main findings and confront them with respect to the scientific literature in a logical and focused fashion.  The main outcomes of the study were not soundly presented.  With respect to the Conclusion section, the authors must highlight the main findings of the study and the possible use of the study outcomes upon buffalo production; conclusions must be aligned with the working hypothesis. The list of references cited in the manuscript (34) is proper, while actualized. This is an interesting study, with a large set of response variables. Yet, on one side the authors must improve both the English language quality as well as the clarity and logical arrangement of the observed results, especially in the Results and Discussion sections. Sorry about this situation but it is necessary to ensure that the paper is readable.  The authors must increase the readability and the scientific writing and merit of the manuscript. Also, the authors must include more detailed information regarding the procedures used in the statistical analyses. All the commented issues and requests should be clearly addressed by the authors; at this point, and based on the above comments, my pronouncement is that this manuscript cannot be accepted in its actual format.  It requires extensive editions and corrections.

Round 2

Reviewer 2 Report

ANIMALS - MDPI, Referee’s Evaluation Report

MANUSCRIPT IDENTIFICATION:  animals – 1598941 – R1

Quercetin alleviates endoplasmic reticulum stress-induced apoptosis in

buffalo ovarian granulosa cells. (ORIGINAL ARTICLE)

Comments to Authors/Editor:

The paper of Yang & colleagues aimed to understand the protective mechanism of quercetin, -a plant-derived flavonoid with anti-oxidant and anti-inflammatory activities-, as a reliever of endoplasmic-reticulum stress, an important cause of granulosa cell apoptosis and, in turn, follicular atresia. This R1-version manuscript was modified according to the suggestions raised by this Reviewer.  The English quality, grammar, and sentence structure were improved, besides, the authors are now following the requested format by the MDPI journals. The Simple Summary was included; yet, I disagree with the idea that Granulosa cells are the most important cells in the follicle; the authors must temper such a statement. Also, as previously suggested, the authors must define any used abbreviation the first time appearing in the manuscript (i.e. L16, L23). L41, … causes in the accumulation …. ??? – nonsense structure. The authors included information regarding buffaloes and their importance in the region. While the objectives of the study were stated, no working hypothesis of the study was proposed; this is a must. In the Material & Methods section, I strongly request more information regarding the reproductive cycle of Buffaloes; the authors never clarified if they are seasonal or non-seasonal reproducers; if so, the authors must indicate what time of the year the experimental samples were collected. Again, the authors must indicate whether the experimental samples were collected at the beginning, the middle, or the last stage of the breeding season. Although the authors defined the number of replicates used in this study, they never mentioned if the samples were collected from homogeneous experimental units??? In general, the materials, standards, and methods used are relevant and in accordance with the objectives of the study. Also, all the sampling techniques, the reported methodology, and consider techniques used in the experiment are detailed and accurate, while in accordance with the objectives of the study, yet, without any single citation. The English quality must be improved also in this section. Again, L184, statistical analysis or statistical analyses???; the authors must include the statistical model used along with the analyses. Since the authors used increased tunicamycin concentrations, the authors must explain why they used a one-way analysis of variance to detect any treatment effect.  Again, the experimental design, and statistical models, were not described well enough for the reader to understand how the experiment was carried out. Besides, the authors must describe if the response variables depicted a normal distribution or if they required an adjustment or transformation in order to perform the analyses through ANOVA. In the Results section; as mentioned in the original revision, do not use the word “significantly” along with the probability level; only use the probability level itself. Regarding the Results section, the novelty value of the results is reasonable. In this R1 version, the authors included some kind of quantitative information aside from the P-values of the observed results regarding the response variables. About the Discussion section, as suggested in the original revision, at the beginning of the Discussion, I do strongly suggest initiating this section including the working hypothesis of the study. Authors must define if, with the obtained results, such a hypothesis is rejected or non-rejected. For this reason, the authors must include the working hypothesis prior to the objectives in the Introduction section. In this R1 version, the authors explained their main findings along with the discussion section and proposed some physiologic explanations for such specific outcomes, considering previous similar studies from the scientific literature. One concern is that since the authors did not consider any scientific hypothesis in the introduction section, from my point of view both the Discussion and the Conclusions sections lack cohesion. Therefore, it seems that the main outcomes of the study are not soundly presented. That is, both Discussion and Conclusions must be aligned with the scientific hypothesis, yet, unfortunately, such a hypothesis never was proposed. This R1 version was significantly improved, yet, the authors must include more detailed information regarding the procedures used in the statistical analyses. All the commented issues and requests should be clearly addressed by the authors; a scientific hypothesis must be included in the study while Discussion and Conclusion must be aligned to such proposed hypothesis.  At this point, and based on the above comments, my pronouncement is that this manuscript requires moderate editions.

Author Response

请参阅附件
